# Composing Task Knowledge with Modular Successor Feature Approximators

**Wilka Carvalho**[*,1]  **Angelos Filos**[2]
**Richard L. Lewis**[1]  **Honglak lee**[1,3]  **Satinder Singh**[1]
[1]University of Michigan  [2]University of Oxford  [3]LG AI Research

## Abstract

Recently, the Successor Features and Generalized Policy Improvement (SF&GPI) framework has been proposed as a method for learning, composing, and transferring predictive knowledge and behavior. SF&GPI works by having an agent learn predictive representations (SFs) that can be combined for transfer to new tasks with GPI. However, to be effective this approach requires state features that are useful to predict, and these state-features are typically hand-designed. In this work, we present a novel neural network architecture, "Modular Successor Feature Approximators" (MSFA), where modules both discover what is useful to predict, and learn their own predictive representations. We show that MSFA is able to better generalize compared to baseline architectures for learning SFs and modular architectures for learning state representations.

## 1 Introduction

Consider a household robot that needs to learn tasks including picking up dirty dishes and cleaning up spills. Now consider that the robot is deployed and encounters a table with both a spill and a set of dirty dishes. Ideally this robot can combine its training behaviors to both clean up the spill and pickup the dirty dishes. We study this aspect of generalization: combining knowledge from multiple tasks.

Combining knowledge from multiple tasks is challenging because it is not clear how to synthesize either the behavioral policies or the value functions learned during training. This challenge is exacerbated when an agent also needs to generalize to novel appearances and environment configurations. Returning to our example, our robot might need to additionally generalize to both novel dirty dishes and to novel arrangements of chairs.

Successor features (SFs) and Generalized Policy Improvement (GPI) provide a mechanism to combine knowledge from multiple training tasks (Barreto et al., 2017; 2020). SFs are predictive representations that estimate how much state-features (known as "cumulants") will be experienced given a behavior. By assuming that reward has a linear relationship between cumulants and a task vector, an agent can efficiently *compute* how much reward it can expect to obtain from a given behavior. If the agent knows multiple behaviors, it can leverage GPI to compute which behavior would provide the most reward (see Figure 2 for an example). However, SF&GPI commonly assume hand-designed cumulants and don't have a mechanism for generalizing to novel environment configurations.

Modular architectures are a promising method for generalizing to distributions outside of the training distribution (Goyal et al., 2019; Madan et al., 2021). Recently, Carvalho et al. (2021a) presented "FARM" and showed that learning multiple state modules enabled generalization to environments with unseen environment parameters (e.g. to larger maps with more objects). In this work, we hypothesize that modules can further be leveraged to discover state-features that are useful to predict.

---

[*]Contact author: wcarvalh@umich.edu.

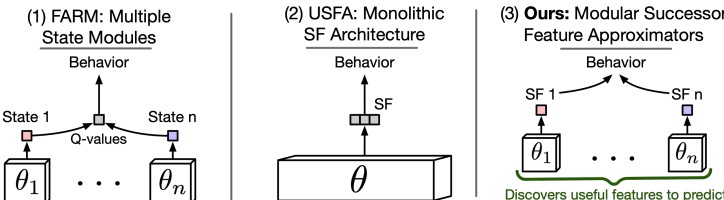

Figure 1: (1) FARM learns multiple state modules. This promotes generalization to novel environments. However, it has no mechanism for combining task solutions. (2) USFA learns a single monolithic architecture for predicting SFs and can combine task solutions. However, it relies on hand-designed state features and has no mechanism for generalization to novel environments. (3) We combine the benefits of both. We leverage modules for reward-driven discovery of state features that are useful to predict. These form the basis of their own predictive representations (SFs) and enables combining task solutions in novel environments.

We present "Modular Successor Feature Approximators" (MSFA), a novel neural network for discovering, composing, and transferring predictive knowledge and behavior via SF&GPI. MSFA is composed of a set of modules, which each learn their own state-features and corresponding predictive representations (SFs). **Our core contribution** is showing that an inductive bias for modularity can enable reward-driven discovery of state-features that are useful for zero-shot transfer with SF&GPI. We exemplify this with a simple state-feature discovery method presented in Barreto et al. (2018) where the dot-product between state-features and a task vector is regressed to environment reward. This method enabled transfer with SF&GPI in a continual learning setting but had limited success in the zero-shot transfer settings we study. While there are other methods for state-feature discovery, they add training complexity with mutual information objectives (Hansen et al., 2019) or meta-gradients (Veeriah et al., 2019). With MSFA, by adding *only an architectural bias for modularity*, we discover state-features that (1) support zero-shot transfer competitive with hand-designed features, and (2) enable zero-shot transfer in visually diverse, procedurally generated environments. We are hopeful that our architectural bias can be leveraged with other discovery methods in future work.

## 2 Related Work on Generalization in RL

**Hierarchical RL** (HRL) is one dominant approach for combining task knowledge. The basic idea is that one can sequentially combine policies in time by having a "meta-policy" that sequentially activates "low-level" policies for protracted periods of time. By leveraging hand-designed or pre-trained low-level policies, one can generalize to longer instructions (Oh et al., 2017; Corona et al., 2020), to new instruction orders (Brooks et al., 2021), and to novel subtask graphs (Sohn et al., 2020; 2022). We differ in that we focus on combining policies *concurrently* in time as opposed to sequentially in time. To do so, we develop a modular neural network for the SF&GPI framework.

**SFs** are predictive representations that represent the current state as a summary of the *successive* features to follow (see §3 for a formal definition). By combining them with Generalized Policy Improvement, researchers have shown that they can transfer behaviors across object navigation tasks (Borsa et al., 2019; Zhang et al., 2017; Zhu et al., 2017), across continuous control tasks (Hunt et al., 2019), and within an HRL framework (Barreto et al., 2019). However, these works tend to require hand-designed cumulants which are cumbersome to design for every new environment. In our work, we integrate SFs with Modular RL to facilitate reward-driven discovery of cumulants and improve successor feature learning.

**Modular RL** (MRL) (Russell & Zimdars, 2003) is a framework for generalization by combining value functions. Early work dates back to (Singh, 1992), who had a mixture-of-experts system select between separately trained value functions. Since then, MRL has been applied to generalize across robotic morphologies (Huang et al., 2020), to novel task-robot combinations (Devin et al., 2017; Haarnoja et al., 2018), and to novel language

instructions (Logeswaran et al., 2021). MSFA, is the first to integrate MRL with SF&GPI. This integration enables combining task solutions in novel environment configurations.

**Generalizing to novel environment configurations with modules**. Goyal et al. (2019) showed that leveraging modules to learn a *state function* improved out-of-distribution generalization. Carvalho et al. (2021a) showed that a modified attention mechanism led to strong generalization improvements with RL. MSFA differs from both in that it employ modules for learning *value functions* in the form of SFs. This enables a principled way to compose task knowledge while additionally generalizing to novel environment configurations.

## 3    Problem Setting and Background

We study a reinforcement learning agent's ability to transfer knowledge between tasks in an environment. During training, the experiences $n_{\texttt{train}}$ tasks $\mathbb{M}_{\texttt{train}} = \{\mathcal{M}_i\}_{i=1}^{n_{\texttt{train}}}$, sampled from a training distribution $p_{\texttt{train}}(\mathcal{M})$. During testing, the agent is evaluated on $n_{\texttt{test}}$ tasks, $\{\mathcal{M}_i\}_{i=1}^{n_{\texttt{test}}}$, sampled from a testing distribution $p_{\texttt{test}}(\mathcal{M})$. Each task $\mathcal{M}_i$ is specified as Partially Observable Markov Decision Process (POMDP), $\mathcal{M}_i = \langle \mathcal{S}^e, \mathcal{A}, \mathcal{X}, R, p, f_x \rangle$. Here, $\mathcal{S}^e$, $\mathcal{A}$ and $\mathcal{X}$ are the environment state, action, and observation spaces. $p(\cdot|s_t^e, a_t)$ specifies the next-state distribution based on taking action $a_t$ in state $s_t^e$, and $f_x(s_t^e)$ maps the underlying environment state to an observation $x_t$. We focus on tasks where rewards are parameterized by a task vector $w$, i.e. $r_t^w = R(s_t^e, a_t, s_{t+1}^e, w)$ is the reward obtained for transition $(s_t^e, a_t, s_{t+1}^e)$ given task vector $w$. Since this is a POMDP, we need to learn a state function that maps histories to agent state representations. We do so with a recurrent function: $s_t = s_\theta(x_t, s_{t-1}, a_{t-1})$. Given this learned state, we want to obtain a behavioral policy $\pi(s_t)$ that best maximises the expected reward it will obtain when taking an action $a_t$ at a state $s_t$: $Q_t^{\pi,w} = Q^{\pi,w}(s_t, a_t) = \mathbb{E}_\pi \left[ \sum_{t=0}^\infty \gamma^t r_t^w \right]$.

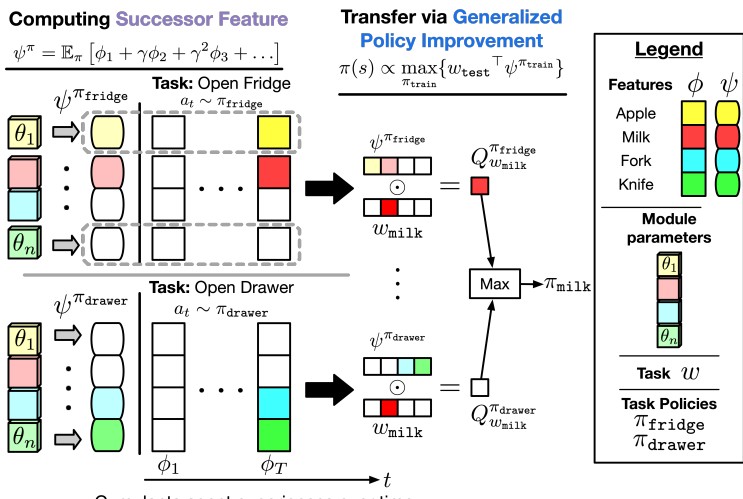

Figure 2: **High-level diagram of how MSFA can be leveraged for transfer with SF&GPI**. During training, we can have the agent learn policies for tasks—e.g. "open drawer" and "open fridge". Each task leads the agent to experience different aspects of the environment—e.g. a "fork" during "open drawer" or an "apple" during "open fridge". We can leverage MSFA to have different modules learn different "cumulants", $\phi$, and SFs, $\psi$. For example, module 1 ($\theta_1$) can estimate SFs for apple cumulants. Module SFs are combined to form the SF for a policy. When the agent wants to transfer its knowledge to a test task—e.g., "get milk"—it can compute Q-values for that task as a dot-product with the SFs of each training task. The highest Q-value is then used to select actions.

**Transfer with SF&GPI.** In order to leverage SFs (Barreto et al., 2017), one assumes an agent has access to state features known as "cumulants", $\phi_t = \phi(s_t, a_t, s_{t+1})$. Given a

behavioral policy $\pi(a|s)$, SFs are a type of value function that use $\phi_t$ as pseudo-rewards:

$$\psi_t^\pi = \psi^\pi(s_t, a_t) = \mathbb{E}_\pi \left[ \sum_{i=0}^\infty \gamma^i \phi_{t+i} \right] \tag{1}$$

If reward is (approximately) $r_t^w = \phi_t^\top w$, then action-values can be decomposed as $Q_t^{\pi,w} = \psi_t^{\pi\top} w$. This is interesting because it provides an easy way to *reuse* task-agnostic features $\psi_t^\pi$ for new tasks.

We can re-use the SFs we've learned from training tasks $\mathbb{M}_{\texttt{train}}$ for transfer with GPI. Assume we have learned (potentially optimal) policies $\{\pi_i\}_{i=1}^{n_{\texttt{train}}}$ and their corresponding SFs $\{\psi^{\pi_i}(s, a)\}_{i=1}^{n_{\texttt{train}}}$. Given a test task $w_{\texttt{test}}$, we can obtain a new policy with GPI in two steps: (1) compute Q-values using the training task SFs (2) select actions using the highest Q-value. This operation is summarized as follows:

$$\pi(s_t; w_{\texttt{test}}) \in \arg\max_{a \in \mathcal{A}} \max_{i \in \{1, \ldots, n_{\texttt{train}}\}} \{Q_t^{\pi_i, w_{\texttt{test}}}\} = \arg\max_{a \in \mathcal{A}} \max_{i \in \{1, \ldots, n_{\texttt{train}}\}} \{\psi_t^{\pi_i\top} w_{\texttt{test}}\} \tag{2}$$

This is useful because the GPI theorem states that $\pi$ will perform as well as all of the training policies, i.e. that $Q^{\pi, w_{\texttt{test}}}(s, a) \geq \max_i Q^{\pi_i, w_i}(s, a) \forall (s, a) \in (\mathcal{S} \times \mathcal{A})$ (Barreto et al., 2017).

SF&GPI enable transfer by exploiting structure in the RL problem: a policy that maximizes a value function is guaranteed to perform at least as well as the policy that defined that value function. However, SF&GPI relies on combining a fixed set of SFs. Another form of transfer comes from "Universal Value Function Approximators" (UVFAs) (Schaul et al., 2015), which add the task-vector $w$ as a parameter to a Q-approximator parameterized by $\theta$, $Q_\theta(s, a, w)$. If $Q_\theta$ is smooth with respect to $w$, then $Q_\theta$ should generalize to test tasks nearby to train tasks in task space. Borsa et al. (2019) showed that one could combine the benefits of both with "Universal Successor Feature Approximators". Since rewards $r^w$, and therefore task vectors $w$, reference deterministic task policies $\pi_w$, one can parameterize successor feature approximators with task-vectors $\tilde{\psi}^{\pi_w} = \tilde{\psi}^w \approx \psi_\theta(s, a, w)$. However, USFA assumed hand-designed cumulants. We introduce an architecture for reward-driven discovery of cumulants and improved function approximation of universal successor features.

## 4  Modular Successor Feature Approximators

We propose a new architecture *Modular Successor Feature Approximators* (MSFA) for approximating SFs, shown in Figure 3. Our hypothesis is that learning cumulants and SFs with modules improves zero-shot composition of task knowledge with SF&GPI. MSFA accomplishes this by learning $n$ state modules $\{s_{\theta_k}\}_{k=1}^n$ that evolve with independent parameters $\theta_k$ and have sparse inter-module information flow. MSFA then produces modular cumulants $\{\tilde{\phi}_t^{(k)}\}_{k=1}^n$ and SFs $\{\tilde{\psi}_t^{\pi,k}\}_{k=1}^n$ by having their computations depend **only** on individual module-states. For example, a cumulant may correspond to information about apples, and would be a function **only** of the module representing state information related to apples. This is in contrast to prior work, which learns a single monolithic prediction module for computing cumulants and SFs (see Figure 3).

The rest of section is structured as follows. In section §4.1, we derive Modular Successor Feature Learning within the Modular RL framework. We then describe our architecture, MSFA, for learning modular successor features in §4.2. In §4.3, we describe how to generate behavior with MSFA. Finally, we describe the learning algorithm for MSFA in section §4.4.

### 4.1  Modular Successor Feature Learning

Following the Modular RL framework (Russell & Zimdars, 2003), we assume that reward has an additive structure $R(s_t, a_t, s_{t+1}) = \sum_k R^k(s_t, a_t, s_{t+1}) = \sum_k R_t^{(k)}$, where $R_t^{(k)}$ is the reward of the $k$-th module. We enforce that every module decomposes reward into an inner-product between its own task-description $w^{(k)}$ and task-agnostic cumulants $\phi^{(k)} \in \mathbb{R}$: $R_t^{(k)} = \phi_t^{(k)} \cdot w^{(k)}$. Here, we simply break up the task vector into $n$ pieces so individual

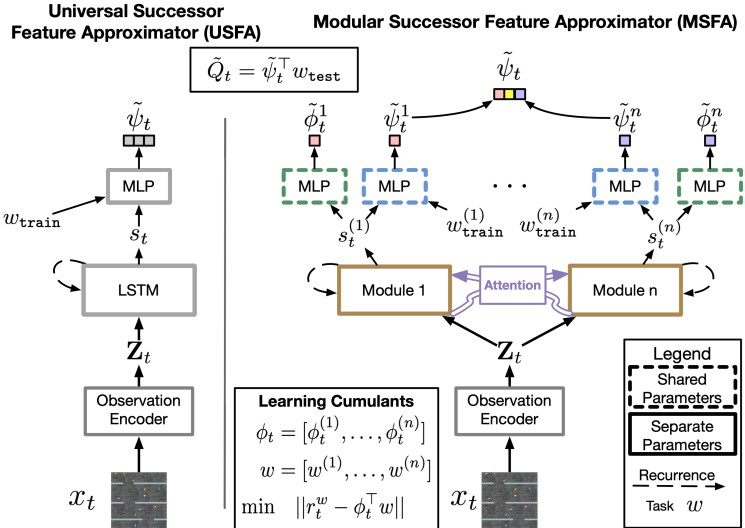

Figure 3: **Left: Universal Successor Feature Approximator (USFA)** learns a single, monolithic successor feature estimator that uses **hand-designed** cumulants. **Right: Modular Successor Feature Approximator (MSFA)** learns a set of successor feature modules, each with their own functions for (a) updating module-state, (b) computing cumulants, and (c) estimating successor features. Modules then share information with an attention mechanism. We hypothesize that isolated module computations facilitate learning cumulants that suppot generalization with GPI.

modules are responsible for subsets of the task vector. This allows us to decompose the action-value function as

$$Q^\pi(s_t, a_t, w) = \sum_{k=1}^{n} Q^{\pi,k}(s_t, a_t, w^{(k)}) = \sum_{k=1}^{n} \psi^{\pi,k}(s_t, a_t) \cdot w^{(k)} \tag{3}$$

where we now have *modular SFs* $\{\psi^{\pi,k}(s,a)\}_{k=1}^{n}$ (see the Appendix for a derivation). Rather than hand-designing modules or cumulants, we aim to discover them from the environment reward signal.

## 4.2 ARCHITECTURE

We learn a set of modules with states $\mathbb{S}_t = \{s_t^{(k)}\}_{k=1}^{n}$. They update at each time-step $t$ with the observation $x_t$, the previous module-state $s_{t-1}^{(k)}$, and information from other modules $A_\theta(s_{t-1}^{(k)}, \mathbb{S}_{t-1})$. Following prior work (Santoro et al., 2018; Goyal et al., 2019; Carvalho et al., 2021a), we have $A_\theta$ combine transformer-style attention (Vaswani et al., 2017) with a gating mechanism (Parisotto et al., 2020) to enforce that inter-module interactions are sparse. Since $A_\theta$ is not the main contribution of this paper, we describe these computations in more detail with our notation in the Appendix. We summarize the high-level update below.

$$s_t^{(k)} = s_{\theta_k}(x_t, s_{t-1}^{(k)}, A_\theta(s_{t-1}^{(k)}, \mathbb{S}_{t-1})) \tag{4}$$

**We learn modular cumulants and SFs** by having sets of cumulants and SFs depend on individual module-states. Module cumulants depend on the module-state from the current and next time-step. Module SFs depend on the current module-state and on their subset of the task description. We summarize this below:

$$\tilde{\phi}_t^{(k)} = \phi_\theta(s_t^{(k)}, a_t, s_{t+1}^{(k)}) \qquad \tilde{\psi}_t^{w,k} = \psi_\theta(s_t^{(k)}, a_t, w^{(k)}) \tag{5}$$

We highlight that cumulants share parameters but differ in their input. This suggests that the key is not having cumulants and SFs with separate parameters but that they are functions

of sparse subsets of state (rather than all state information). We show evidence for this hypothesis in Figure 6.

We concatenate module-specific cumulants and SFs to form the final outputs: $\tilde{\phi}_t = \left[\tilde{\phi}_t^{(1)}, \ldots, \tilde{\phi}_t^{(n)}\right]$ and $\tilde{\psi}_t^w = \psi_\theta(s_t, a_t, w) = \left[\tilde{\psi}_t^{w,1}, \ldots, \tilde{\psi}_t^{w,n}\right]$. Note that cumulants, are only used during learning, update with the module-state from the next time-step.

### 4.3 BEHAVIOR

During **training**, actions are selected in proportion to Q-values computed using task SFs as $\pi(s_t, w) \propto \tilde{Q}(s_t, a, w) = \psi_\theta(s_t, a, w)^\top w$. In practice we use an epsilon-greedy policy, though one can use other choices. During **testing**, we compute policies with GPI as $\pi(s_t, w_{\texttt{test}}) \in \arg\max_a \max_{z \in \mathbb{M}_{\texttt{train}}} \{\psi_\theta(s_t, a, z)^\top w_{\texttt{test}}\}$, where $\mathbb{M}_{\texttt{train}}$ are train task vectors.

### 4.4 LEARNING ALGORITHM

MSFA relies on three losses. The first loss, $\mathcal{L}_Q$, is a standard Q-learning loss, which MSFA uses to learn optimal policies for the training tasks. The main difference here is that MSFA uses a particular parameterization of the Q-function $Q^{\pi_w, w}(s, a) = \psi^{\pi_w}(s, a)^\top w$. The second loss, $\mathcal{L}_\psi$, is an SF learning loss, which we use as a regularizer to enforce that the Q-values follow the structure in the reward function $r_t^w = \phi_t^\top w$. For this, we again apply standard Q-learning but using SFs as value functions and cumulants as pseudo-rewards. The final loss, $\mathcal{L}_\phi$ is a loss for learning cumulants that grounds them in the environment reward signal. The losses are summarised as follows

$$\mathcal{L}_Q = ||r_t + \gamma\psi_\theta(s_{t+1}, a', w)^\top w - \psi_\theta(s_t, a_t, w)^\top w||^2 \tag{6}$$

$$\mathcal{L}_\psi = ||\tilde{\phi}_t + \gamma\psi_\theta(s_{t+1}, a', w) - \psi_\theta(s_t, a_t, w)||^2 \tag{7}$$

$$\mathcal{L}_\phi = ||r_t^w - \tilde{\phi}_t^\top w||^2 \tag{8}$$

where $a' = \arg\max_a \psi_\theta(s_{t+1}, a, w)^\top w$. Selecting the next action via the combination of all modules ensures they individually convergence to optimal values (Russell & Zimdars, 2003). The final loss is $\mathcal{L} = \alpha_Q \mathcal{L}_Q + \alpha_\psi \mathcal{L}_\psi + \alpha_\phi \mathcal{L}_\phi$.

## 5 EXPERIMENTS

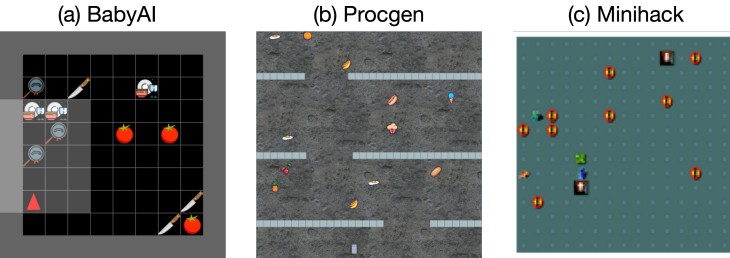

Figure 4: **We study an agent's ability to combine task knowledge in three environments.** (a) In BabyAI, an agent learns to pick up one object type at a time during training. During testing, the agent must pickup combinations of object types while avoiding other object types. This is the setting used by USFA which assumed **hand-designed** cumulants. (b) In Procgen, we study extending this form of generalization to a visually diverse, procedurally generated environment. (c) In Minihack, we go beyond combining object navigation skills. Here, an agents needs to combine (1) avoiding teleportation traps, (2) avoiding monsters, and (3) partial visibility around the agent.

We study generalization when training behaviors must be combined concurrently in time in the presence of novel object appearances and layouts. The need to combine training behaviors

tests how well MSFA can leverage SF&GPI. Generalization to novel object appearances and layouts tests how well MSFA's modular construction supports generalization to novel environment configurations.

**Baselines.** (1) **Universal Value Function Approximator (UVFA)** (Schaul et al., 2015), which takes the task as input: $Q_\theta(s, w)$. This comparison shows shows the transfer benefits of SF&GPI. (2) **UVFA with Feature-Attending Recurrent Modules (UVFA+FARM)** instead takes state-factors as input $Q_\theta(s^{(1)}, \ldots, s^{(n)}, w)$. Each state-factor $s^{(k)}$ is the output of a FARM module. (3) **Modular Value Function Approximator(MVFA)** is an adaptation of (Haarnoja et al., 2018) where modules learn individual Q-values $Q_\theta^{(i)}(s^{(i)}, w^{(i)})$. Comparing to UVFA+FARM and MVFA enables us to study the benefits of leveraging modules for learning value functions in the form of SFs. (4) **Universal Successor Function Approximator (USFA)** (Borsa et al., 2019) leverages a single monolithic function for successor features with **hand-designed cumulants**. USFA is an upper-bound baseline that allows us to test the quality of cumulants and successor features that MSFA learns. We also test a variant of USFA with learned cumulants, **USFA-Learned-$\phi$**, which shows how the architecture degrades without oracle cumulants.

**Implementation.** We implement the state modules of MSFA with FARM modules (Carvalho et al., 2021b). For UVFA and USFA, we learn a state representation with an LSTM (Hochreiter & Schmidhuber, 1997). We implement all $\phi$, $\psi$, and $Q$ functions with Mutli-layer Perceptrons. We train UVFA and UVFA+FARM with n-step Q-learning (Watkins & Dayan, 1992). When learning cumulants, USFA and MSFA have the exact same losses and learning alogirthm. They both learn Q-values and SFs with n-step Q-learning. We use $n = 5$. When not learning cumulants, following (Borsa et al., 2019), USFA only learns SFs with n-step Q-learning. All agents are built with JAX (Bradbury et al., 2018) using the open-source ACME codebase (Hoffman et al., 2020) for reinforcement learning.

## 5.1 Combining object navigation task knowledge

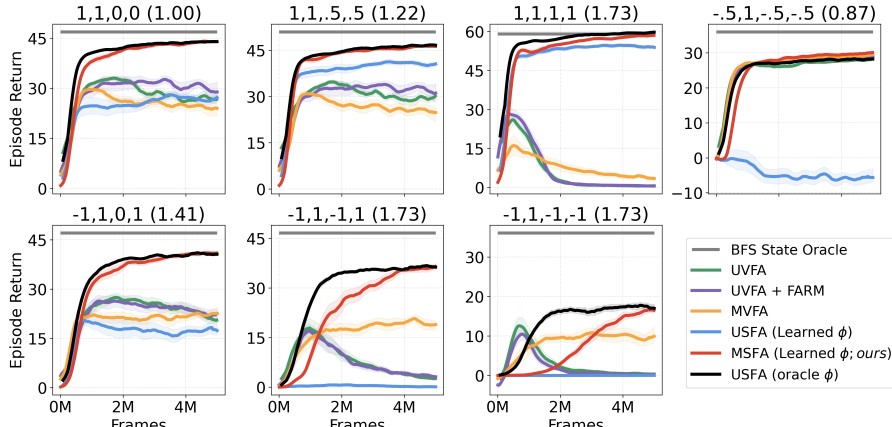

Figure 5: **MSFA matches USFA, which has hand-designed cumulants**. We show mean and standard error generalization episode return across 10 runs. We put a task's L2 distance to the closest train task in parenthesis. USFA best generalizes to novel combinations of picking up and avoiding objects. Once USFA learns cumulants, its performance degrades significantly. UVFA-based methods struggle as more objects should be avoided or tasks are further in distance to train tasks.

MSFA learns modular functions for computing $\phi$ and estimating $\psi$. We hypothesize that this facilitates learning cumulants that respond to different aspects of the environment (e.g. to different object categories). This leads to the following research questions. **R1**: Can we recover prior generalization results that relied on hand-designed cumulants for different object categories? **R2**: How important is it to learn modular functions for $\phi$ and $\psi$? **R3**: Without GPI, do learning modular functions for $\phi$ and $\psi$ still aid in generalization?

**Setup.** We implement a simplified version of the object navigation task of (Borsa et al., 2019) in the (Chevalier-Boisvert et al., 2019) BabyAI environment. The environment consists of 3 instances of 4 object categories. **Observations** are partial and egocentric. **Actions**: the agent can rotate left or right, move forward, or pickup an object. When it picks up an object, following (Borsa et al., 2019), the object is respawned somewhere on the grid. **Task** vectors lie in $w \in \mathbb{R}^4$ with training tasks being the standard unit vectors. For example, $[0, 1, 0, 0]$ specifies the agent must obtain objects of category 2. **Generalization** tasks are linear combinations of training tasks. For example, $[-1, 1, -1, 1]$ specifies the agent must obtain categories 2 and 4 while *avoiding* categories 1 and 3. Borsa et al. (2019) showed that USFA could generalize with *hand-designed* cumulants that described whether an object was picked up. We describe challenges for this task in detail in the Appendix.

**R1: MSFA is competitive with USFA, which uses oracle $\phi$.** Figure 5 shows USFA with a similar generalization trend to (Borsa et al., 2019). Tasks get more challenging as they are further from train tasks or involve avoiding more objects. For simply going to combinations of objects, USFA-Learned-$\phi$ does slightly worse than MSFA. However, with more objects to avoid, all methods except MSFA (including USFA-Learned-$\phi$) degrades significantly. For comparison, we show performance by an oracle bread-first-search policy with access to ground-truth state (**BFS State Oracle**). All methods have room for improvement when objects must be avoided. In the appendix, we present heat-maps for how often object categories were picked up during different tasks. We find that MSFA most matches USFA, while USFA-Learned-$\phi$ commonly picks up all objects regardless of task.

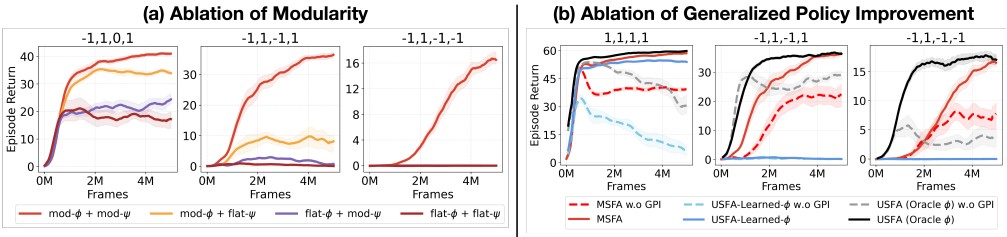

Figure 6: **Learning modular $\phi$ and $\psi$ is key to generalization and improves generalization even without GPI.** We show mean and standard error generalization episode return across 10 runs. (a) We ablate having modular functions for $\phi_\theta$ and $\psi_\theta$. Generalization results degrade significantly. (b) We ablate leveraging GPI for generalization from all SF-based methods. MSFA without GPI can outperform both USFA-Learned-$\phi$ with GPI and USFA without GPI. This shows the utility of modularity for generalization.

**R2: Learning modular $\phi$ and $\psi$ functions is critical for generalization.** Learning an entangled function corresponds to learning a monolithic function for $\psi$ or $\phi$ where we concatenate module-states, e.g. $\tilde{\psi}_t^w = \psi_\theta(s_t^{(1)}, \ldots, s_t^{(n)}, a, w)$. Modular functions correspond to equation 5. Figure 6 (a) shows that without modular functions for $\phi$ **and** $\psi$, performance severely degrades. This also highlights that a naive combination of USFA+FARM—with entangled functions for $\phi$ and $\psi$—does not recover our generalization performance.

**R3: Modularity alone improves generalization.** For all SF-based methods, we remove GPI and select actions with a greedy policy: $\pi(s) = \arg\max_a \psi_\theta(s_t, a, w)^\top w$. Figure 6 (b) shows that GPI is critical for generalization with USFA as expected. USFA-Learned-$\phi$ benefits less from GPI (presumably because of challenges in learning $\phi$). Interestingly, MSFA can generalize relatively well without GPI, sometimes doing better than USFA without GPI.

## 5.2 COMBINING OBJECT NAVIGATION TASK KNOWLEDGE WITH NOVEL APPEARANCES AND ENVIRONMENT CONFIGURATIONS

Beyond generalizing to combinations of tasks, agents will need to generalize to different layouts and appearances of objects. **R4**: Can MSFA enable combining task knowledge in a visually diverse, procedurally generated environment?

**Setup.** We leverage the "Fruitbot" environment within ProcGen (Cobbe et al., 2020). Here, an agent controls a paddle that tries to obtain certain categories of objects while avoiding

others. When the agent hits a wall or fence, it dies and the episode terminates. If the agent collects a non-task object, nothing happens. **Observations** are partial. **Actions**: At each time-step the agent moves one step forward and can move left or right or shoot pellets to open fences. **Training and generalization tasks** follow the same setup as §5.1.

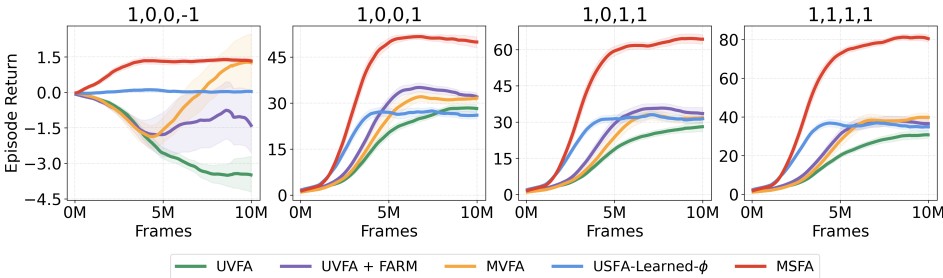

Figure 7: **MSFA is able to combine task knowledge in a visually diverse, procedurally generated ProcGen environment.** We find that no method is able to do well when there are objects to avoid ($w = [1, 0, 0, -1]$) in this setting (see text for more). However, as more objects need to be collected MSFA best generalizes (10 runs).

**R4: MSFA enables combining task knowledge of object navigation tasks in a visually diverse, procedurally generated environment.** Figure 7 shows that when an agent has to generalize to collecting more objects, modular architectures generalize best, with MSFA doing best. When objects have to be avoided, we see that no architecture does well, though MSFA tends to do better. We observe that avoiding objects leads agents to hit walls. Since the agent always moves forward at each time-step in Fruibot, this makes avoiding objects a particularly challenging type of generalization.

## 6 DISCUSSION AND CONCLUSION

We have presented "Modular Successor Feature Approximators", a modular neural network for learning cumulants and SFs produced by their own modules. We first showed that MSFA is competitive with prior object navigation generalization results that relied on hand-designed cumulants (§5.1). Afterwards, we showed that MSFA improves an agent's ability to combine task knowledge in a visually diverse, procedurally generated environment (§5.2). We also show that MSFA can combine solutions of heterogeneous tasks (§B). Our ablations show that learning modular cumulants and SFs is critical for generalization with GPI.

We compared MSFA to (1) USFA, a monolithic architecture for learning cumulants and SFs; (2) FARM, an architecture which learns multiple state-modules but combines them with a monolithic Q-value function. Our results show that when learning cumulants, MSFA improves generalization with SF&GPI compared to USFA. Additionally, without GPI, MSFA as an architecture improves generalization as compared to both FARM and USFA.

**Limitations**. While we demonstrated reward-driven discovery of cumulants for transfer with SF&GPI, we focused on relatively simple task encodings. Future work can extend this to more expressive encodings such as language embeddings. Another limitation is that we did not explore more sophisticated state-feature discovery methods such as meta-gradients (Veeriah et al., 2019). Nonetheless, we think that MSFA provides an important insight for future work: that modularity is a simple but powerful inductive bias for discovering state-features that enable zero-shot transfer with SF&GPI.

**Future directions**. SFs are useful for exploration (Janz et al., 2019; Machado et al., 2020); for discovering and combining options (Barreto et al., 2019; Hansen et al., 2019); for transferring policies across environments (Zhang et al., 2017); for improving importance sampling (Fujimoto et al., 2021); and for learning policies from other agents (Filos et al., 2021). We hope that future work can leverage MSFA for improved state-feature discovery and SF-learning in all of these settings.

## Acknowledgments

This work was supported in part by a University of Michigan Rackham Merit Fellowship and grant from LG AI Research. The authors would also like to thank Andrew Lampinen, Cameron Allen, and the anonymous reviewers for their helpful comments.

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
