# OpenReview forum: "Composing Task Knowledge With Modular Successor Feature Approximators"
_ICLR.cc/2023/Conference — ICLR 2023 poster_

### Official Review · Reviewer_H7je · 2022-10-15

**Confidence:** 4
**Correctness:** 4
**Technical Novelty And Significance:** 3
**Empirical Novelty And Significance:** 3
**Recommendation:** 6

**Clarity, Quality, Novelty And Reproducibility:**

The paper is written clearly, and is easy to follow. Authors did a good job in explaining their motives, method, and results. Regarding the reproducibility, nothing has been mentioned about publicly sharing the code.

**Strength And Weaknesses:**

The idea behind MSFA which is to learn disentangled cumulants and successor features is very interesting and promising. Overall, the paper is theoretically and practically reasonable. One thing that is worth studying is how modules in the MSFA architecture are correlated with one another.

In the experiments section, it is not mentioned how many seeds were used to obtain the provided results. Clarification regarding that would be useful. On a similar note, according to results provided, 4 runs were used. In order to better validate the generalizability and robustness of the MSFA, running more tests are required.

**Summary Of The Paper:**

In this paper authors propose a method to generalize agents better by combining knowledge from multiple tasks. While the agent should generalize on a single given task, it should be able to do so across multiple ones. The proposed method, MSFA, aims at solving this problem.

**Summary Of The Review:**

See above.

---

> ### Author Response · Authors · 2022-11-18
> **Thank you for the review. Experiments are now with 10 runs and we added anonymous github**
>
> We are glad that you found the paper well-written and clear to understand. Below we respond to your questions:
>
> **How many seeds?** All experiments involved 4 independent runs. We have run 6 more independent runs (10 total) for all experiments and updated the results.
>
> **Reproducibility.** We have created an anonymous repository to enable other researchers to build off of our work:
>
> https://anonymous.4open.science/r/msfa-6188/README.md
>
>
> **Correlation**. We have added an analysis of the correlation between modules in Appendix E, Figure 12. Overall, we see that modules are weakly correlated with each other.

---

### Official Review · Reviewer_xfgM · 2022-10-23

**Confidence:** 4
**Correctness:** 4
**Technical Novelty And Significance:** 2
**Empirical Novelty And Significance:** 3
**Recommendation:** 6

**Clarity, Quality, Novelty And Reproducibility:**

Paper is well-written and has high-quality. While the approach is essentially the mix of two existing research, it has good amount of contribution built in. The main drawback of the paper is the reproducibility as details of the experimentation are not present.

Minor:
- monolothic => Did you mean monolithic?

**Strength And Weaknesses:**

Strengths
+ Paper is well-written with solid contribution in the space
+ Generalization across tasks is a critical step towards practical reinforcement learning. Hence it is of great interest to the community
+ Ablation studies are thorough showing the impact of modularity and generalization even in heterogeneous tasks

Weaknesses
- Reproducibility: code or algorithm were not discussed in the paper. While few parameters were covered, it is unlikely for someone to be able to reproduce the results.
- The empirical results are done in toy domains. What are the main limits to apply MSFA to larger and more realistic tasks?

**Summary Of The Paper:**

Authors tackled the problem of combining knowledge from multiple tasks. They introduced Modular Successor Feature Approximator (MSFA) as a new network architecture that both identifies what is important to predict and also learns the corresponding representation. From another lens MSFA removes the need to hand-design cumulants (state-dependent features that capture the return linearly) for the Universal Successor Feature Approximator work by Borsa et al. [ICLR-2019].


**Summary Of The Review:**

Good submission with yellow flag on reproducibility.

---

> ### Author Response · Authors · 2022-11-18
> **Thank you for the review. We added code via anonymous repo**
>
> Thank you for your review. We are glad that you find this to be a good submission. Below we respond to specific questions:
>
> ### Yellow flag for reproducibility:
>
> We have created an anonymous repository to enable other researchers to build off of our work:
>
> https://anonymous.4open.science/r/msfa-6188/README.md
>
>
>
> ### Limitations to applying MSFA to larger, more realistic tasks
>
> Generalizing across more complex tasks may involve combining more complex behaviors. This may benefit from more expressive task representations. In this work, like prior SF&GPI work [1,2], we trained on unit vectors and transferred across their span. This is effectively transferring to conjunctions and negations of the training tasks. For more complex tasks, one can benefit from a distributed task representation $w$ produced by a neural network. In this setting, $\phi$ is also distributed since $r=\phi^{\top}w$. For example apple and tomato features would be represented across multiple (likely overlapping) dimensions in $\phi$ and $w$. Recent work on vision and language models such as CLIP [3], indicates that a dot-product between observations features $\phi$ and language encodings $w$ can be a powerful vehicle for zero-shot generalization. The main challenge would be in jointly learning $w$ and $\phi$ to be useful fo SF&GPI. This is currently an open problem in the literature. Despite, we believe that the insights of this paper--that having $\phi^{(i)}$ and $\psi^{(i)}$  be sparse functions of state information improves zero-shot composition of task knowledge---will be helpful in enabling generalization with a richer task space.
>
> Please let us know if there is another challenge posed by larger, more realistic tasks which you are would like us to discuss.
>
> ### Regarding results being in toy domains
>
> We want to highlight the relative complexity of Minihack and FruitBot, which are both procedurally generated environments. Both of these environments are more complex than BabyAI: they have more complex visual backgrounds, more cluttered scenes, and changing object appearances. This challenges required that we made the following changes to all architectures or could not learn the tasks:
>
> - Small Atari CNN [4] --> a ResNet
> - Uniform Experience Replay --> Prioritized Experience Replay
> - Increasing Capacity of Value Function Approximations: (i.e. hidden sizes changed from [128] to [512, 512]).
>
> The MSFA architecture was otherwise unmodified as we went from the most simple (BabyAI) to the most complex (Minihack and Procgen) environments. These results suggest that MSFA has the potential to scale with visual complexity if one incorporates environment-centric best practices for scaling the architectural and algorithmic pieces (e.g., larger networks, prioritized experience replay, etc.).
>
> [1] Barreto et al, Transfer in Deep Reinforcement Learning Using Successor Features and Generalized Policy Improvement, ICML, 2018
>
> [2] Borsa et al, Universal Successor Feature Approximators, ICLR, 2019
>
> [3] Radford et al, Learning Transferable Visual Models From Natural Language Supervision, ICML, 2021.
>
> [4] Minh et al, Human-level Control through Deep Reinforcement Learning, Nature, 2013

---

### Official Review · Reviewer_GLtt · 2022-10-24

**Confidence:** 4
**Correctness:** 3
**Technical Novelty And Significance:** 3
**Empirical Novelty And Significance:** 3
**Recommendation:** 8

**Clarity, Quality, Novelty And Reproducibility:**

The paper is mostly clear to read, has high quality, is fairly novel, and is decently reproducible.

**Strength And Weaknesses:**

Strengths

- Well motivated problem
- Simple proposed method
- Succinct results

Weaknesses

- Parts of writing can be improved
- Experiments are performed on a limited number of (simple/toy-like) domains

**Summary Of The Paper:**

The paper studies the problem of task generalization through the use of the successor features framework. It proposes using a modular architecture with the universal successor features method (USFA; introduced in prior work). The modular architecture enforces disentanglement by conditioning each module's predictions on only a subset of the other modules. The experiments show that such a modular architecture is indeed useful in improved task generalization when combined with a USFA algorithm variant which learns the cumulants.

**Summary Of The Review:**

I really like how succinctly the paper conveys the main message (precisely Figure 5 and Figure 6 are enough to convince the efficacy of the proposed modular architecture). The paper is written well (barring a few confusions I have below around some of the definitions) and is simple to understand for the most part. Overall, I think it makes a useful contribution, with well-constructed experiments and clear writing. Hence, I vote for acceptance.

In Figure 3, $z_t$ is the output of the encoder, which, going by the text, should be denoted by $s_t$?
Why is $x_t$ provided as input in Eq 4? Shouldn’t $s_t$ already encode that information?
A_theta is defined as a function of $s^k_t$ and $S_{t-1}$, but wouldn’t $S_{t-1}$ already include $s^k_t$, given how it’s defined in the first line of Section 4.2?

---

> ### Author Response · Authors · 2022-11-18
> **Thank you for the review**
>
> Thank you for the positive review. We are happy you appreciated this submission. Below we respond in detail to your questions.
>
> * $z\_t$ is a latent representation produced by an observation encoder and $s^{(k)}\_t$ is the output of each individual state-module. We originally omitted $s^{(k)}\_t$ to reduce clutter. We have updated the figure to display $s^{(k)}\_t$ to improve clarity.
> * $x\_t$ is input to Eq4 because we assume a partially observable environment and the agent needs to learn its own representation for state $\mathbb{S}\_t$, which is learned with state modules. We accidentally used the same variable $s\_t$ for both agent state, which the agent learns, and for the environment state, which is unknown. We have updated the text to use $\mathcal{S}^e\_t$ and $s^e\_t$ to refer to the set of environment states and a single environment state, respectively.
> * $A\_{\theta}$ should be in terms of $s^{(k)}\_{t-1}$ not $s^{(k)}\_t$. A module’s previous module-state $s^{(k)}\_{t-1}$ is used to select information from the set of all module states $\mathcal{S}\_t$. Thank you for pointing this out—we have clarified this in the text.

---

### Official Review · Reviewer_kJTC · 2022-10-24

**Confidence:** 4
**Correctness:** 1
**Technical Novelty And Significance:** 2
**Empirical Novelty And Significance:** 2
**Recommendation:** 3

**Clarity, Quality, Novelty And Reproducibility:**

The paper is well written with lots of intuitions provided for why the proposed architectural bias and modularity is important; however, there seem to be broad claims that are not well justified. The title is rather misleading, and it is not clear to me how compositional aspects of the task is actually learnt to then help for generalization? The architecture consists of attention modules, which have been studied previously as well; and simply the addition of this does not well justify the broad claims in my opinion.


**Strength And Weaknesses:**


	- The main idea of this paper is to propose a new architecture that can achieve modularization, leading to better generalization. The paper achieves this by learning useful representations from such a modular architecture, in experiments with procedurally generated environments.
	- The paper considers a train test generalization setting where the agent is trained on a seqeuence of tasks to then test for generalization to new tasks.
	- The key idea is to use attention mechanism in the resulting architecture that can capture different aspects of the task. Given observations, and the encoded reprsentation Z, I want to understand more why these separate modules are actually useful?
	- I understand the intuition behind the work, but how these modules are actually capturing what the paper claims to achieve is not clear? What are the different components of Z_t that each of the module is trying to capture? Why would simply using an attention module be useful here? I am not sure how the intuitions are justified from this architecture?
	- The paper claims to achieve disentangled cumulants? How do you justify that? Is there a simple example demonstrating this?
	- Experimental results compare to existing baselines such as UVFA. However, there are other architectural modular networks such as RIMS (Goyal et al) that also proposed such architectures? Why are these baselines architectural changes not considered?
	- My biggest worry of this type of work is the claims the authors try to make, but the results are only in terms of returns or some existing evaluation metrics, which does not justify the claims? The paper provides lots of intuitions of why the architectural bias is important, in terms of disentangling cumulants in the context of successor features; where the architecture captures different aspects of the tasks that helps for generalization. However, empirical results do not fully support this, neither are there theoretical claims for it.
	- Why is the the term compositional exactly used here? How is this term justified? I think the paper makes a broad claim but fails to fully backup results for it


**Summary Of The Paper:**

This paper proposes a new architecture that is useful for generalization to new tasks, that may be previous combination of prior tasks. It adapts from existing literature on successor features and GPI, and shows that a modified architecture that predicts useful representations while also learning representations can be significant in terms of generalization. This work is an adaptation of recent line of works on modular architectures to achieve OOD generalization.

Overall, there seems to broad claims that the proposed architecture learns compositional aspects of the tasks that helps with generalization to out of distribution tasks.


**Summary Of The Review:**

I would argue for a rejection of this work mainly because of the broad claims in terms of "disentanglement" and "compositionality" which are not well justified from the results. The paper proposes a simple archiectural change, variations of which has been proposed in the past well. As such, the proposed work is not fully novel, yet there are broad claims which makes the paper look really fancy!

---

> ### Author Response · Authors · 2022-11-18
> **Thank you for the constructive feedback. Response Part 1**
>
>
> We thank the reviewer for their constructive feedback. We would like to acknowledge their concerns about how we phrased our contributions in terms of disentanglement and compositional generalization. Below we explain changes in nomenclature we've made in response to this review and rationalize some of our original decisions. We emphasize that we are open to modifying our language further.
>
>
>
> ### Clarifying the main claim of the paper
>
> MSFA is an architecture for learning $\phi$ and $\psi$  to use with the SF&GPI framework. We re-iterate that prior work has hand-designed $\phi$, which is cumbersome to do for every environment, or learned $\phi$ that had limited success in the zero-shot transfer settings we study. **Our core contribution is showing that leveraging modules to estimate their own cumulants and successor features improves zero-shot composition of task knowledge with Successor Features and Generalized Policy Improvement.**  We verify that estimating $\phi$ and $\psi$ with individual modules is critical for combining task knowledge with SF&GPI in Figure 6 and extend these results to visually complex environments.
>
>
>
> ### Regarding “compositional task generalization”
>
> We want to begin by acknowledging that there are multiples forms of "composition" that are possible in the machine learning literature. For example, one can combine words in a language setting or attributes in a vision setting.
>
> **We focus on composing task knowledge** in the form of value functions. Specifically, we assume reward functions $r(s,a)$ that are linear in task encoding $w$, i.e.  $r^{\tt train}\_i (s,a) = \phi(s,a)^\top w^{\tt train}\_i$.
> Our training task knowledge is successor features (SFs) we learn $\psi\_{\pi\_i}(s,a)$ for $m$ training tasks $w^{\\tt train}\_i$, which individually estimate $\psi\_\pi(s,a) = \mathbb{E}\_\pi\left[\phi\_t + \gamma\phi\_{t+1} + \ldots \right]$. We combine task knowledge using the SF&GPI framework to generalize to test tasks with rewards $r^{\tt test}(s,a) = \phi(s,a)^\top w^{\tt test} = \phi(s,a)^\top \left(\sum\_i\alpha\_i w^{\tt train}\_i \right)$.
>
> **We did not intend to make claims** that (1) MSFA discovers compositional aspects of the task or (2) that MSFA discovers different aspects of the tasks. On the contrary, we train the agent on "basis tasks" that enable generalizing across the space of task combinations. We changed the caption of Figure 2 to not suggest that MSFA discover compositional aspects of tasks. We would appreciate it if you could let us know of any other parts that suggest this.
>
> We want to emphasize that if you feel there is more appropriate phrasing, we are happy to put it in the revision.
>
>
>
> ### Regarding “disentanglement”, we will instead use “modular”:
>
> We originally chose “disentangled function” to indicate that MSFA used functions for producing cumulants that only considered state information from their own modules. To be more precise, we have modules learn a set of $n$ module-states $\\{s^{i}\\}^{n=1}\_i$, which collectively make up the agent's state representation. Consider producing cumulants with this state representation using a linear function/transformation $A$. We call the following an "entangled function"/"monolithic function":
>
> $$
> \begin{bmatrix}
>          \phi^1 \\\\
>          \vdots \\\\
>          \phi^n
>  \end{bmatrix}= \begin{bmatrix}
>          A\_{11} & \cdots & A\_{1m}\\\\
>          \vdots & A\_{22} & \vdots\\\\
>          A\_{n1} & \cdots & A\_{nm}
>  \end{bmatrix} \begin{bmatrix}
>          s^1 \\\\
>          \vdots \\\\
>          s^n
>  \end{bmatrix}
> $$
>
> By "entangled," we mean that a cumulant $\phi^i$ is (potentially) a function of all module-state information  $\phi^i=\sum\_j A\_{ij} s^j$. We did not mean to confuse the reader with other notions of disentanglement (e.g. disentangled factors of variation) so we have replaced all usage of the word “disentangled” with “modular”. By a "modular" or "disentangled" function, we simply meant that we were learning a function like the following
>
> $$\begin{bmatrix}
>          \\phi^1 \\\\
>          \\vdots \\\\
>          \\phi^n
>  \\end{bmatrix} = \begin{bmatrix}
>          A\_{1} & 0 & 0\\\\
>          0 & A\_{2} & 0\\\\
>          0 & 0 & A\_{n}
>  \end{bmatrix}\begin{bmatrix}
>          s^1 \\\\
>          \vdots \\\\
>          s^n
>  \end{bmatrix}$$
>
> In our setting, we had $A\_1=\ldots=A\_n$ be a shared MLP and each $s^i$ was produced by a module with its own parameters. Our experiments demonstrate leveraging modules to learn $\phi^{(i)}$ *and* $\psi^{(i)}$ is an effective way to obtain zero-shot combination of task knowledge with the SF&GPI framework. Figure 6 directly tests this hypothesis. We hope this alleviates your concerns that our experiments do not support this hypothesis.
>
> (edit: fixed error in math rendering that weirdly worked before)

---

> > ### Author Response · Authors · 2022-11-18
> > **Part 2**
> >
> > ### Regarding the title:
> >
> > We have changed it to “Composing Task Knowledge with Modular Successor Feature Approximators”.
> >
> >
> >
> > ### Regarding novelty of the architecture:
> >
> > We want to clarify that RIMS aims to learn modules for representing state $s\_t$. We differ in that we aim to learn modules for representing value functions (SFs) $\psi\_\pi^{(i)}(s,a)$ over their own cumulants $\phi^{(i)}(s,a)$. By doing so, the SF&GPI framework provides a principled way to combine training task knowledge. In Figure 6 (b) we show that, even without GPI, this inductive bias of combining successor features produced by their own modules can improve zero-shot generalization over both a monolithic architecture (USFA-Learned-$\phi$) and an architecture that learns a modular state function (UVFA-FARM). This is a novel integration of predictive state representations (SFs) and modular architectures. To the best of our knowledge, no work has integrated the two before. We have clarified this in the text.
> >
> >
> >
> > ### Regarding having RIMs as a baseline
> >
> > We want to highlight that we compared against Feature-Attending Recurrent Modules (FARM) via UVFA-FARM. FARM [1] is a direct descendent of RIMs which augmented the attention-mechanism used by RIMs for stronger generalization results in RL. This is why we chose FARM over RIMs. We have clarified this in the text.
> >
> > [1] Carvalho et al, Feature-Attending Recurrent Modules for Generalization in Reinforcement Learning, NeurIPS Deep RL Workshop, 2022
> >
> >
> >
> > ### We have added another baseline studying modularity in the value function
> >
> > We have added a Modular Value Function Approximator baseline  (MVFA) by adapting [2] to our setting. MVFA studies modularity for learning $Q(s,a)$. Like [1], it has modules produce Q-values which are then summed $Q(s,a) = \sum\_i Q^{(i)}(s,a)$ . We implement MVFA by having individual modules only predict $Q^{(i)}$ instead of $\phi^{(i)}$ and $\psi^{(i)}$. This allows us to study the importance of having modules learn SFs and combine task knowledge with SF&GPI. Our experiments show that is generalizes relatively well but not nearly as well as MSFA.
> >
> > [2] Haarnoja et al, Composable Deep Reinforcement Learning for Robotic Manipulation (2018)
> >
> >
> >
> > ### Visualization of the modules in a simple setting.
> >
> > We have added a visualization for our first experiment in Appendix D. Here, we followed the same experimental setup as [3, 4]. State-transitions where an agent picks up an object category are rewarded. Each object category defines one task and tasks are defined as the standard unit vectors. E.g. $[1,0,0,0]$ indicates that the agent should pickup objects of category A. We test generalization to combinations of task encodings. In this setting, modules should activate in response to individual object categories. We present results in Appendix D confirming this. In this setting, it is not surprising that individual modules can learn to represent individual object categories. What is potentially more surprising is that having SFs $\psi^{(i)}$ be functions of just their own module-state $s^{(i)}$ is critical for our zero-shot transfer performance---if we don't leverage modularity for both $\phi$ and $\psi$, we do not recover our performance. We highlight that [3] had limited zero-shot transfer success here, especially in transfer settings with more objects to avoid, whereas we perform as well as [4] which uses oracle $\phi$.
> >
> > [3] Barreto et al, Transfer in Deep Reinforcement Learning Using Successor Features and Generalized Policy Improvement, ICML, 2018
> >
> > [4] Borsa et al, Universal Successor Feature Approximators, ICLR, 2019

---

### Author Response · Authors · 2022-11-19
**Joint response to all authors**

We thank the reviewers for their very constructive feedback which we have used to improve the paper. Overall, we are encouraged that the reviewers found this to be a high quality submission studying a well-motivated and important problem [RGLtt(2), RxfgM(3)]; that they found we present a useful and interesting contribution with well constructed experiments [RGLtt(2), RxfgM(3), RH7je(4)]; and that they found the paper over well-written [RkJTC(1), RGLtt(2), RxfgM(3), RH7je(4)] with clearly explained motives, methods, and results [RGLtt(2), RxfgM(3), RH7je(4)]. We respond to individual comments below but summarize overall changes we made to the paper here.

**Summary of changes**:

* We have created an anonymous github repository for the code: https://anonymous.4open.science/r/msfa-6188/README.md
* We have updated the submission files. All changes are in red.
* All results previously reported used 4 independent runs. All results now use 10 independent runs.
* We have added a baseline "Modular Universal Value Function Approximator (MVFA)" which leverages modules for *only* learning the action-value functions $Q(s,a)$ (i.e. modules learn $Q^{(i)}_{\theta}$). This tests the importance of leveraging modules for learning value functions in the form of Successor Features.

---

### Decision · Program_Chairs · 2023-01-20

**Decision:**

Accept: poster

**Justification For Why Not Higher Score:**

More exhaustive experiments would be needed for a spotlight presentation.

**Justification For Why Not Lower Score:**

The main concern from the most negative reviewer was well addressed in the rebuttal. Unfortunately, the corresponding reviewer did not react to the rebuttal and left his score unchanged.

**Metareview: Summary, Strengths And Weaknesses:**

The paper presents a new approach for task generalization using  successor features.A modular architecture is used with the universal successor features, where the modular architecture enforces disentanglement of the feature vectors. The experiments show an improved task generalization.

Strength: The paper presents a well motivated idea for task generalization which is also intuitive. The results are presented very well and convey the improved task generalization abilities. All reviewers appreciated that the paper contains interesting ideas. On the downside, concerns were raised about the complexity and number of experimental setups where the approach has been evaluated. Yet, these concerns were alleviated mostly in the rebuttal. Moreover, Reviewer kJTC was concerned about several claims in the paper that they are not backed up by empirical results. I think these concerns were also addressed well in the rebuttal. Unfortunately reviewer kJTC chose to not join the discussion and did not update his score. However, due to his missing response I give his review less weight.


**Note From Pc:**

if the above contains the word "oral" or "spotlight" please see: "oral" presentation means -> notable-top-5% and "spotlight" means -> notable-top-25%. As stated in our emails, we are disassociating presentation type from AC recommendations

**Summary Of Ac-Reviewer Meeting:**

N/A